# Maternal knowledge, attitudes and practices related to neonatal jaundice and associated factors in Shenzhen, China: a facility-based cross-sectional study

Ying Huang,[1,2] Ling Chen,[1] Xiaojiao Wang,[1] Chun Zhao,[1] Zonglian Guo,[3] Jue Li,[1] Fang Yang,[4] Wenzhi Cai  [1,2]

YH and LC contributed equally.

YH and LC are joint first authors.

For numbered affiliations see end of article.

**Correspondence to**
Professor Wenzhi Cai;
caiwzh@smu.edu.cn

## ABSTRACT

**Objective** This study aimed to assess knowledge, attitudes and practices related to neonatal jaundice among mothers in Shenzhen, China, and analyse associated factors.

**Design** A cross-sectional study.

**Setting** This study was conducted in Shenzhen Hospital, Southern Medical University, a university-affiliated, tertiary level A, public hospital in China. On average, 4000 mothers are discharged from this hospital after childbirth each year, most of whom can access a mobile phone and the internet.

**Participants** Participants were 403 mothers discharged from the study hospital within 48–72 hours after vaginal delivery or 96–120 hours after caesarean delivery between April and June 2021. Participants were recruited using convenience sampling.

**Primary outcome** Mothers' knowledge, attitudes and practices related to neonatal jaundice, modelled using binary logistic regression.

**Secondary outcomes** Factors associated with mothers' knowledge, attitudes and practices related to neonatal jaundice.

**Results** The questionnaire was reliable (Cronbach's alpha=0.802) and valid (scale-level content validity index=0.958). The valid response rate was 96.4%. Only 46.4% of participating mothers had good knowledge about neonatal jaundice and 41.7% indicated they would seek information about neonatal jaundice. A binary logistic regression analysis showed good knowledge about jaundice was associated with a high education level (ie, master's degree or above; OR=5.977, 95% CI: 1.994 to 17.916, p=0.001), prior education on neonatal jaundice (OR=3.617, 95% CI: 1.637 to 7.993, p=0.001) and male babies (OR=1.714, 95% CI: 1.122 to 2.617, p=0.013). A positive attitude toward jaundice was associated with being cared for by a 'yuesao' (maternity matron specialised in caring for mothers and newborns) (OR=1.969, 95% CI: 1.264 to 3.066, p=0.003) and good knowledge about jaundice (OR=1.804, 95% CI: 1.194 to 2.726, p=0.005). Finally, good practices related to neonatal jaundice were associated with prior education on neonatal jaundice (OR=2.260, 95% CI: 1.105 to 4.625, p=0.026)

and good knowledge about jaundice (OR=3.112, 95% CI: 2.040 to 4.749, p<0.001).

**Conclusion** Many mothers have poor knowledge about jaundice, especially regarding causes, danger signs and breast milk jaundice. Maternal information-seeking behaviour about neonatal jaundice needs to be improved. Medical staff should incorporate information about the causes/danger signs of jaundice and breast milk jaundice in maternal health education. It is also necessary to strengthen health education for mothers, especially those with low education and no yuesao, and provide reliable websites where mothers can obtain information about neonatal jaundice.

## STRENGTHS AND LIMITATIONS OF THIS STUDY

⇒ A strength of this study was that participants were mothers with healthy infants who had been discharged from the birth hospital; these mothers may be more likely to ignore the problem of jaundice.

⇒ The investigation time was the peak period of jaundice occurrence, which aimed to avoid recall bias.

⇒ Recruiting mothers and then surveying them at a later time may have prompted mothers to search for information about jaundice.

⇒ This study focused on new mothers and did not include significant others who may have roles in care of the newborn (eg, grandmothers, sisters or aunts).

## INTRODUCTION

Neonatal jaundice, also known as neonatal hyperbilirubinemia, refers to yellow staining of the skin or other organs caused by the accumulation of bilirubin in the body.[1] It is a common clinical problem in the neonatal period, and approximately 50%–60% of full-term infants and 80% of premature infants develop jaundice within 1 week after birth.[2] In many infants, neonatal jaundice is a benign condition. However, severe hyperbilirubinemia may cause acute bilirubin

encephalopathy (ABE) or kernicterus, which may progress to nerve deafness, choreoathetoid cerebral palsy, intellectual disability and even death.[3] [4] A report from China showed that from January to December 2009, 348 cases of kernicterus were recorded in 33 hospitals.[5] In addition, a worldwide survey reported that at least 480 700 newborns develop severe hyperbilirubinemia each year, with the risk for kernicterus being 13% (n=75 400) and that for death being 24% (n=114 100).[6] Therefore, neonatal jaundice is a serious threat to the life and health of newborns, and the associated high rates of disability and mortality place heavy burdens on society and families.

Early detection and timely treatment of neonatal jaundice are key strategies to prevent ABE and kernicterus. However, neonatal jaundice generally peaks on the 5th–7th day after birth,[1] at which time most healthy full-term infants have been discharged from hospital. Therefore, most neonatal jaundice occurs at home. As the main caregivers of newborns after discharge from hospital, mothers are often the first to observe jaundice, its progression and early signs of ABE and kernicterus. They are central to achieving favourable outcomes for management of neonatal jaundice. Wennberg et al[7] reported that providing mothers with detailed information about neonatal jaundice and its risks was associated with a reduction in the incidence of ABE in Nigeria. The Stop Kernicterus in Nigeria consortium[8] demonstrated that a delay in seeking care, regardless of birth site, was a major contributor to ABE and kernicterus and proposed that maternal education on neonatal jaundice should be targeted as an intervention strategy. The clinical practice guidelines for neonatal jaundice from the National Institute for Health and Care Excellence[9] and American Academy of Pediatrics Subcommittee[2] also recommend parents and caregivers are educated about neonatal jaundice, especially on how to check their baby for jaundice and what to do when jaundice is suspected. These guidelines suggest that maternal jaundice instruction be given high priority among the myriad topics. However, effective instruction starts with meaningful engagement between hospital staff and mothers.[10] Hospital staff therefore need to clarify what mothers know about jaundice and their current attitudes and practices, which will allow health education programmes to target identified gaps. Although similar investigations have been done in other countries or regions including Ghana,[11] Accra[12] and Egypt,[13] no evaluation tools or reports related to maternal knowledge, attitudes and practices about neonatal jaundice are available in China.

Therefore, this study designed a questionnaire to assess knowledge, attitudes and practices related to neonatal jaundice among mothers in Shenzhen, China. The information obtained maybe used to assist healthcare providers in designing educational programmes to improve awareness about neonatal jaundice among mothers, which will ultimately help to prevent disability and deaths among newborns.

## METHODS

### Study setting and design
We conducted an anonymous, self-administered, cross-sectional survey involving 403 mothers in Shenzhen, China from April to June 2021. Participants were recruited using convenience sampling.

### Study population
The target population was mothers who gave birth at Shenzhen Hospital, Southern Medical University, China from April to June 2021.

### Inclusion and exclusion criteria
Mothers were eligible for this study if they were discharged from the hospital without serious illness after childbirth and had access to a mobile phone and the internet. Mothers were excluded if they were not the main caregiver of their newborn after being discharged from hospital or could not complete the questionnaire by themselves. Moreover, we excluded mothers whose newborns were admitted to the neonatal intensive care unit for treatment or that died after birth.

### Data collection tools
Data were collected using a questionnaire that covered baseline characteristics, knowledge, attitudes and practices regarding neonatal jaundice. These items were developed with reference to: (1) an established guideline on neonatal jaundice,[9] (2) an integrative review[14] and (3) three services for investigating maternal knowledge, attitude and behaviour regarding neonatal jaundice.[13] [15] [16] We also consulted seven neonatologists and discussed the items among the research team. Following the review by the expert panel, nine mothers whose babies had experienced neonatal jaundice were conveniently recruited to provide input on the importance and clarity of the questionnaire items. Mothers were asked to suggest alternative wording for existing items and identify items that required deletion and addition as necessary. Some items were modified based on this review. For example, we modified 'have you ever learned about neonatal jaundice' to 'prior health education on neonatal jaundice', 'G6PD deficiency' to 'broad bean disease' and 'serum total bilirubin is the gold standard for diagnosing neonatal jaundice' to 'blood test is the gold standard for diagnosing neonatal jaundice'. No items were identified for deletion or addition. The draft questionnaire was then pretested with 20 mothers who were conveniently selected from the same hospital based on the study inclusion criteria and exclusion criteria. The final version of the questionnaire included 27 items. Sixteen items evaluated knowledge (categorical responses: 'true', 'false', 'do not know'), seven items assessed attitude (categorical responses: 'strongly agree', 'agree', 'not sure', 'disagree', 'strongly disagree') and four items covered practices (yes/no responses).

### Validity and reliability
The content validity of the questionnaire was appraised quantitatively by sending the final version of the

questionnaire to a group of experts including seven specialists in the field of neonatology. Based on the experts' feedback about relevance, the instrument's scale-level content validity index was calculated as 0.958, which was higher than the value of 0.8 that indicates adequate validity.[17]

Cronbach's alpha[18] was used to check the reliability of the questionnaire, which is the most common measure of internal consistency. In this study, the Cronbach's alphas were 0.802 for the whole questionnaire, 0.789 for the knowledge dimension, 0.721 for the attitude dimension and 0.414 for the practice dimension. The Cronbach's alphas for the knowledge and attitude dimensions were higher than 0.7, and were within the acceptable recommended range. However, that for practice (0.414) was below 0.7.[19] The small number of items (four items) on that dimension may explain the low alpha estimate. To avoid the impact of a small number of items, Cronbach[20] proposed the mean interitem correlation (ρ) in 1951 to estimate the internal consistency of dimensions with a small number of items. Generally, a mean interitem correlation (ρ) between 0.15 and 0.20 indicates acceptable internal consistency.[21] In this study, the mean interitem correlation was 0.15 for the practice dimension, which was within the acceptable range.

### Data collection procedure
On the day each mother was discharged after delivery, the investigator informed them of the purpose, duration and method of this study and obtained their phone number. About 5 days after discharge from hospital, the investigator sent the questionnaire link, which was developed using the 'Question star' platform, to participating mothers via mobile phone and then called the participant to complete the questionnaire on the same day. In total, 413 questionnaires were completed. We excluded 10 participants: 3 did not complete the questionnaire, 3 wrote their names instead of their age, 2 had missing data for age, 1 wrote her age as '240' and 1 participant chose the first option for each question. This left 403 valid questionnaires for analysis, giving an effective recovery rate of 96.4%. Detailed data collection procedures are presented in online supplemental figure 1.

### Data processing and analysis
We used SPSS V.25.0 (IBM Corp) for all data analyses. Descriptive statistics were calculated for baseline characteristics and categorical variables using simple frequencies and percentages. The main outcome variables were knowledge, attitudes and practices. The knowledge score for each participant was determined by allotting a score of '1' to correct responses and '0' (zero) to incorrect and 'do not know' responses. Therefore, the maximum obtainable knowledge score was 16. A knowledge score≤10 was considered poor knowledge, and scores>10 represented good knowledge.[22] The attitude scores were obtained by assigning points to responses on the 5-point Likert scale (1 point for 'strongly disagree' to 5 points for 'strongly

agree'). The maximum obtainable attitude score was 35 points. An attitude score≤28 was considered a poor attitude and scores>29 were categorised as good attitudes.[22] Similarly, the practice score for each participant was determined by allotting a score of '1' to correct responses and '0' (zero) to incorrect responses. This gave a maximum obtainable practice score of 4. Those with a practice score<4 were regarded as having poor practice, whereas a score of 4 was considered to reflect good practice.[22] Finally, the levels of knowledge, attitude and practice were coded as 0 for poor, 1 for good.[22] Chi-square tests (bivariable analyses) were used to determine the associations between the baseline and outcome variables. All variables with p<0.05 in the bivariable analysis were included in the binary logistic regression (multivariable analysis) to determine the associations between dependent (knowledge, attitudes and practices) and independent (baseline characteristics) variables. In consideration of having yuesao or not is related to salary range and education, which are supposed to affect attitude and knowledge towarding to neonatal jaundice. Thus, another comparison had been done to differentiate those without yuesao and with yuesao in correlation to education, salary in terms of attitude and knowledge. Statistical significance was represented by p<0.05 at a 95% CI.

### Patient and public involvement
Before the formal survey, the researchers interviewed 20 mothers to determine the readability of the questionnaire, the time required and the best way to conduct the investigation.

## RESULTS
### Participants' baseline characteristics
Of the 403 mothers included in our analyses, a majority were aged 28–32 years (48.9%) and 75.7% had a spontaneous vaginal delivery. Most mothers (80.6%) had received prior health education about jaundice from health workers on the day of normal discharge from the delivery hospital (48–72 hours after vaginal delivery or 96–120 hours after caesarean delivery). Among the 403 participating mothers, 113 (28%) reported their current child was admitted to the hospital for treatment due to jaundice after discharge, and 56 (13.9%) had a previous child with a history of neonatal jaundice. Participants' baseline characteristics are presented in table 1.

### Mothers' knowledge of neonatal jaundice
We found that 45.4% of participating mothers had good knowledge regarding neonatal jaundice. The rates of correct answers to the knowledge questions ranged from 29% to 96.8%. Questions that showed low rates of correct answers were: 'cranial haematoma may cause neonatal jaundice' (29%), 'blood test is the gold standard for diagnosing neonatal jaundice' (34.2%), 'it is abnormal for jaundice to appear within 24 hours after birth' (36.2%), 'it is abnormal for jaundice to reappear after it subsides' (37%) and 'breast milk jaundice is a benign

**Table 1** Baseline characteristics of participating mothers (N=403)

| Variables | Characteristics | n (%) |
|---|---|---|
| **Sociodemographic data** | | |
| Age, years | 19–27 | 114 (28.3) |
| | 28–32 | 197 (48.9) |
| | 33–45 | 92 (22.8) |
| Blood group | O | 145 (36.0) |
| | A | 121 (30.0) |
| | B | 106 (26.3) |
| | AB | 31 (7.7) |
| Education level | High school and below | 68 (16.9) |
| | University | 310 (76.9) |
| | Postgraduate and above | 25 (6.2) |
| Occupation | Employed | 267 (66.3) |
| | Self-employed | 40 (9.9) |
| | Homemaker | 82 (20.3) |
| | Others | 14 (3.5) |
| Average family monthly income, yen | ≤5000 | 50 (12.4) |
| | 5001–10 000 | 154 (38.2) |
| | 10 001–20 000 | 125 (31) |
| | 20 001–30 000 | 35 (8.7) |
| | ≥30 001 | 39 (9.7) |
| Time from the place of residence to the delivery hospital, min | ≤10 | 52 (12.9) |
| | 10–30 | 212 (52.6) |
| | 30–60 | 126 (31.3) |
| | ≥60 | 13 (3.2) |
| **Delivery history** | | |
| Parity | Primipara | 224 (55.6) |
| | Multipara | 179 (44.4) |
| Delivery mode | Spontaneous vaginal | 305 (75.7) |
| | Caesarean section | 98 (24.3) |
| **Infant's information** | | |
| Sex | Male | 210 (52.1) |
| | Female | 193 (47.9) |
| Birth weight* | Low | 365 (7.7) |
| | Normal | 31 (90.6) |
| | Hight | 7 (1.7) |
| Feeding method | Exclusive breast feeding | 196 (48.6) |
| | Mixed feeding | 197 (48.9) |
| | Exclusive formula-feeding | 10 (2.5) |
| Cranial haematoma† | Yes | 15 (3.7) |
| | No | 364 (90.3) |
| | Not sure | 24 (6.0) |

Continued

**Table 1** Continued

| Variables | Characteristics | n (%) |
|---|---|---|
| Whether meconium passed within 24 hours | Yes | 397 (98.5) |
| | No | 6 (1.5) |
| Predischarge bilirubin level | Normal | 312 (77.4) |
| | Height | 91 (22.6) |
| 'Yuesao'‡ | Yes | 138 (34.2) |
| | No | 265 (65.8) |
| **Previous experience/exposure to neonatal jaundice** | | |
| Prior health education on neonatal jaundice | Yes | 361 (80.6) |
| | No | 42 (10.4) |
| Family history/friends with neonatal jaundice history (N=373) | Yes | 45 (12.1) |
| | No | 328 (87.9) |
| Previous child with a history of neonatal jaundice | Yes | 56 (13.9) |
| | No | 347 (86.1) |
| Current child admitted to hospital for treatment for jaundice after discharge | Yes | 113 (28.0) |
| | No | 290 (72.0) |
| Mother's knowledge level | Good | 183 (45.4) |
| | Poor | 220 (54.6) |
| Mother's attitude level | Good | 170 (42.2) |
| | Poor | 233 (57.8) |
| Mother's practice level | Good | 214 (53.1) |
| | Poor | 183 (46.9) |

*Weight: low weight<2500 g; normal weight 2500–4000 g; high weight>4000 g.
†Cranial haematoma: haematoma caused by rupture and bleeding of subperiosteal vessels in the parieto-occipital region due to birth injury.
‡Yuesao: maternity matron specialised in caring for mothers and newborns.

and self-limited condition, and interruption of breast feeding is not recommended as a therapeutic intervention' (37%). Table 2 presents scores for knowledge about neonatal jaundice among participating mothers.

### Mothers' attitudes towards neonatal jaundice

The results revealed that 42.2% of participating mothers had poor attitudes towards neonatal jaundice. Over half of the participants strongly agreed that postdischarge observation was necessary and postpartum visits were needed to measure the bilirubin level (52.4% and 51.9%,

**Table 2** Maternal knowledge about neonatal jaundice (N=403)

| Items | True n (%) | False n (%) | Don't know n (%) | Correct rate n (%) |
|---|---|---|---|---|
| **Observation of neonatal jaundice** | | | | |
| When newborns develop jaundice, their skin will turn yellow. | 390 (96.8) | 4 (1.0) | 9 (2.2) | 390 (96.8) |
| When newborns develop jaundice, their face will turn yellow first. | 344 (85.4) | 12 (3.0) | 47 (11.7) | 344 (85.4) |
| When looking for jaundice, check the naked baby in bright and preferably natural light. | 365 (90.6) | 10 (2.5) | 28 (6.9) | 365 (90.6) |
| **Classification of neonatal jaundice** | | | | |
| Neonatal jaundice is divided into physiological jaundice and pathological jaundice. | 371 (92.1) | 3 (0.7) | 29 (7.2) | 371 (92.1) |
| **Danger signs of neonatal jaundice** | | | | |
| Palms and soles turn yellow, indicating that jaundice is severe. | 265 (65.8) | 29 (7.2) | 109 (27.0) | 265 (65.8) |
| It is an abnormal condition if the jaundice appears within first 24 hours. | 146 (36.2) | 155 (38.5) | 102 (25.3) | 146 (36.2) |
| It is an abnormal condition that the jaundice reappears after it has subsided. | 149 (37.0) | 143 (35.5) | 111 (27.5) | 149 (37.0) |
| **Complication of neonatal jaundice** | | | | |
| Severe jaundice may lead to brain damage. | 325 (80.6) | 7 (1.7) | 71 (17.6) | 325 (80.6) |
| **Cause of neonatal jaundice** | | | | |
| The mother's blood type is O, and the father's blood type is A, B or AB, which may cause neonatal jaundice. | 228 (56.6) | 38 (9.4) | 137 (34.0) | 228 (56.6) |
| Cranial haematoma may cause neonatal jaundice. | 117 (29.0) | 39 (9.7) | 247 (61.3) | 117 (29.0) |
| Bowel obstruction may cause neonatal jaundice. | 281 (69.7) | 14 (3.5) | 108 (26.8) | 281 (69.7) |
| Broad bean disease (G6PD) may cause jaundice. | 163 (40.4) | 25 (6.2) | 215 (53.3) | 163 (40.4) |
| Breast feeding may cause jaundice | 200 (49.6) | 109 (27.0) | 94 (23.3) | 200 (49.6) |
| **Breast milk jaundice** | | | | |
| Breast milk jaundice is a benign and self-limited condition, and interruption of breast feeding is not recommended as a therapeutic intervention. | 149 (37.0) | 135 (33.5) | 119 (29.5) | 149 (37.0) |
| **Diagnosis of neonatal jaundice** | | | | |
| Blood test is the gold standard for diagnosing neonatal jaundice. | 138 (34.2) | 127 (31.5) | 138 (34.2) | 138 (34.2) |
| **Treatment of neonatal jaundice** | | | | |
| Phototherapy is a common, effective and safe treatment method for neonatal jaundice. | 367 (91.1) | 3 (0.7) | 33 (8.2) | 367 (91.1) |

respectively). However, some mothers believed that neonatal jaundice was a common physiological phenomenon that would not cause serious consequences (10.9%), and 29.2% lacked understanding that adequate feeding was conducive to resolving jaundice. Mothers' attitudes towards neonatal jaundice are presented in table 3.

### Mothers' practices regarding neonatal jaundice

In general, 53.1% of mothers had good practices regarding neonatal jaundice, and 96% checked their baby for jaundice after discharge. However, only 41.9% indicated they would take the initiative to learn about neonatal jaundice after discharge. Mothers' practices regarding jaundice are shown in table 4.

### Factors associated with knowledge, attitudes and practices related to neonatal jaundice among mothers

The binary logistic regression analysis revealed that good knowledge about jaundice was associated with a high level of education (master's degree or above; OR=5.977, 95% CI: 1.994 to 17.916, p=0.001), receiving prior health education on neonatal jaundice (OR=3.617, 95% CI: 1.637 to 7.993, p=0.001) and male babies (OR=1.714, 95% CI: 1.122 to 2.617, p=0.013). A positive attitude towards jaundice was associated with being cared for by a 'yuesao' (matron specialised in maternal and newborn care) (OR=1.969, 95% CI: 1.264 to 3.066, p=0.003) and good knowledge about jaundice (OR=1.804, 95% CI: 1.194 to 2.726, p=0.005). Finally, good practices related

**Table 3** Maternal attitudes towards neonatal jaundice (N=403)

| Items | Strongly disagree n (%) | Disagree n (%) | Not sure n (%) | Agree n (%) | Strongly agree n (%) |
|---|---|---|---|---|---|
| I think neonatal jaundice is a common physiological phenomenon and will not cause serious consequences.* | 80 (19.1) | 216 (53.6) | 63 (15.6) | 39 (9.7) | 5 (1.2) |
| I think that a baby with jaundice, does not need treatment and will self-recover.* | 151 (38.5) | 201 (49.9) | 43 (10.7) | 6 (1.5) | 2 (0.5) |
| I think it is very important to observe neonatal jaundice after discharged from the hospital. | 5 (1.2) | 0 (0.0) | 6 (1.5) | 181 (44.9) | 211 (52.4) |
| I think it is necessary for postpartum visitors to assess jaundice condition. | 8 (1.0) | 0 (0.0) | 8 (2.0) | 182 (45.2) | 209 (51.9) |
| I think a baby with suspected jaundice should go to a medical institution or community healthcare centre to measure the bilirubin level in a timely manner. | 3 (0.7) | 5 (1.2) | 10 (2.5) | 230 (57.1) | 155 (37.5) |
| I believe that adequate breast feeding is good for jaundice. | 5 (1.2) | 29 (7.2) | 84 (20.8) | 196 (48.6) | 89 (22.1) |
| I think it is necessary to learn knowledge of neonatal jaundice. | 2 (0.5) | 0 (0.0) | 10 (2.5) | 216 (53.6) | 175 (43.4) |

*Reverse scored items.

to jaundice were associated with prior health education on neonatal jaundice (OR=2.260, 95% CI: 1.105 to 4.625, p=0.026) and good knowledge about jaundice (OR=3.112, 95% CI: 2.040 to 4.749, p<0.001). The results of $\chi^2$ tests (bivariable analyses) for maternal knowledge, attitudes and practices related to neonatal jaundice are shown in online supplemental table 1. Factors associated with knowledge, attitudes and practices related to neonatal jaundice among mothers are shown in table 5. Binary logistic regression analysis of maternal knowledge, attitudes and practices related to neonatal jaundice for mothers with and without yuesao are shown in online supplemental table 2 and online supplemental table 3.

## DISCUSSION

In this study, 45.4% of mothers had good knowledge about neonatal jaundice. This knowledge level was better than that reported in other countries such as Nepal[23] (22%), Egypt[24] (30%) and Karbala city, Iraq[25] (34%). This may be attributed to the high proportion (83.1%) of our respondents who had graduated from university. Our study also found that education level was significantly associated with knowledge about neonatal jaundice, which was consistent with the results of an earlier review.[26] Due to differences in scoring algorithms or items investigating attitudes and practices,[23–25] it was difficult to compare attitudes and behaviours reported in our study with those from other countries.

The majority (80.6%) of participants in this study had received prior health education on neonatal jaundice, which suggested that their knowledge about neonatal jaundice would be high. However, this study found that many mothers had poor knowledge regarding neonatal jaundice, with only 45.5% of participating mothers had good knowledge about neonatal jaundice. This large gap may be attributable to the gap of nearly a week between the time they received health education and the time of our investigation; some mothers might have forgotten the content of the health education. In addition, medical staff only provided post-discharge monitoring and follow-up instruction and did not include neonatal

**Table 4** Maternal practices regarding neonatal jaundice (N=403)

| Items | Yes n (%) | No n (%) |
|---|---|---|
| I took the initiative to seek information about neonatal jaundice. | 168 (41.7) | 235 (58.3) |
| After discharge, I checked my infant for jaundice, such as the colour of the skin, sclerae, urine, bowel movements. | 387 (96.0) | 16 (4.0) |
| After discharge, I followed the doctor's instructions to take the infant to a medical institution or community healthcare centre to measure the bilirubin level. | 376 (93.3) | 27 (6.7) |
| After discharge, I breastfed adequately. | 359 (89.1) | 44 (10.9) |

**Table 5** Binary logistic regression analysis of maternal knowledge, attitudes and practices related to neonatal jaundice (N=403)

| Variables | Classification | Knowledge OR | 95%CI | P value | Attitude OR | 95%CI | P value | Practices OR | 95%CI | P value |
|---|---|---|---|---|---|---|---|---|---|---|
| Education level | High school and below (Ref.) | NI | | **0.001** | NI | | | NI | | |
| | College and undergraduate course | 3.011 | 1.563 to 5.800 | 0.001 | | | | | | |
| | Postgraduate student or above | 5.977 | 1.994 to 17.916 | 0.001 | | | | | | |
| Average family monthly income (yen) | ≤5000 (Ref.) | NI | | | NI | | | NI | | |
| | 5001 to 10000 | | | | | | | | | |
| | 10 001 to 20000 | | | | | | | | | |
| | 20 001 to 30000 | | | | | | | | | |
| | ≥30001 | | | | | | | | | |
| Occupation | Employed (Ref.) | NI | | 0.110 | NI | | | NI | | |
| | Self-employed | 0.403 | 0.183 to 0.887 | 0.024 | | | | | | |
| | Homemaker | 0.730 | 0.409 to 1.302 | 0.286 | | | | | | |
| | Other | 1.266 | 0.401 to 3.994 | 0.688 | | | | | | |
| Parts | Primipara (Ref.) | NI | | | NI | | | NI | | |
| | Multipara | | | | 1.498 | 0.995 to 2.255 | 0.053 | | | |
| Prior education on neonatal jaundice | No (Ref.) | NI | | | NI | | | NI | | |
| | Yes | 3.617 | 1.637 to 7.993 | **0.001** | | | | 2.260 | 1.105 to 4.625 | **0.026** |
| Predischarge bilirubin level | Normal | NI | | | NI | | | NI | | |
| | Hight | | | | | | | | | |
| Neonate sex | Female (Ref.) | NI | | | NI | | | NI | | |
| | Male | 1.714 | 1.122 to 2.617 | **0.013** | | | | | | |
| Cranial haematoma | No (Ref.) | NI | | 0.073 | NI | | | NI | | |
| | Yes | 1.549 | 0.510 to 4.706 | 0.440 | | | | | | |
| | It is not clear | 0.321 | 0.112 to 0.920 | 0.034 | | | | | | |

Continued

**Table 5** Continued

| Variables | Classification | Knowledge | | | Attitude | | | Practices | | |
|---|---|---|---|---|---|---|---|---|---|---|
| | | OR | 95%CI | P value | OR | 95%CI | P value | OR | 95%CI | P value |
| Yuesao | No (Ref.) | NI | | | NI | | | NI | | |
| | Yes | | | | 1.969 | 1.264 to 3.066 | 0.003 | | | |
| Knowledge level | Poor (Ref.) | NA | | | NA | | | | | |
| | Good | | | | 1.804 | 1.194 to 2.726 | 0.005 | 3.112 | 2.040 to 4.749 | **0.000** |
| Attitude level | Poor (Ref.) | NI | | | NA | | | | | |
| | Good | | | | | | | 1.498 | 0.983 to 2.283 | 0.060 |

Bold values denote statistical significance to the p<0.05 level.
NA, not applicable; NI, not included in the final logistic regression analysis.

jaundice knowledge related to the questionnaire when conducting health education. Another factor that might have contributed to the comparative ineffectiveness of postnatal instruction was that the unique environment was absent that mothers received health education about jaundice from health workers in a single setting, which combined a lecture, demonstration and interactive discussion, as there is generally a rush to discharge mothers from birthing centres.

The present study reported that many mothers had poor knowledge regarding the causes and danger signs of jaundice. Relatively few mothers provided correct answers to some items; for example, 'cranial haematoma may cause neonatal jaundice' (29%), 'blood test is the gold standard for diagnosing neonatal jaundice' (34.2%), 'it is abnormal for jaundice to appear within 24 hours after birth' (36.2%) and 'it is abnormal for jaundice to reappear after it subsides' (37%). Poor knowledge about the causes and danger signs of jaundice may mean mothers turn to traditional treatments, which results in delays in seeking medical attention for neonatal jaundice, thereby contributing to the development of ABE and kernicterus.[11] Therefore, it is recommended that medical staff incorporate information about the causes and danger signs of jaundice into neonatal jaundice health education programmes. For breast milk jaundice, this involves monitoring the jaundice without changing in the infant's breast feeding;[27] however, 73% of mothers in this study did not know that breast milk jaundice is generally a benign condition, where interruption of breast feeding is not recommended as a therapeutic intervention. Having poor knowledge of breast milk jaundice may mean mothers discontinue breast feeding after jaundice occurs. However brief, such discontinuation may jeopardise an infant's ability to return to exclusive breast feeding, which is unnecessarily harmful to the infant and traumatic for mothers.[28] This means mothers of affected infants should be educated about breast milk jaundice and informed that breast feeding should be interrupted in rare instances (ie, if the neonate displays signs of ABE). In addition, the incidence of G6PD deficiency is high in Shenzhen.[29 30] Traditional Chinese medicine is widely used in China to prevent or treat neonatal jaundice.[31 32] However, neonates with G6PD deficiency that use such remedies may have severe jaundice.[33] Our findings suggested that most mothers (59.5%) did not know the G6PD deficiency may cause jaundice. Therefore, mothers whose neonates have G6PD deficiency should be educated about avoiding using traditional Chinese medicine to treat or prevent neonatal jaundice.

Importantly, our study highlighted that maternal information-seeking behaviour related to neonatal jaundice needs to be improved. Only 41.7% of mothers in this study indicated they took the initiative to seek information about neonatal jaundice. This poor practice could be because physical and psychological changes in the postpartum period mean mothers' energy is limited. However, it could also be attributable to health literacy, which

has an impact on people's health information-seeking behaviour.[34] A previous study[34] reported that the lower the parents' health literacy, the less likely they were to take the initiative to obtain information about their child's health. Actively understanding relevant knowledge will help to improve maternal awareness of neonatal jaundice, which will be conducive to managing neonatal jaundice after discharge from hospital. In addition, with the popularisation of the internet and smart phones, more parents are using these ways to access parenting knowledge. However, they complain that they face major challenges in identifying whether the information is reliable.[35 36] Therefore, it is recommended that when providing education about neonatal jaundice, medical staff also provide mothers with some reliable websites to facilitate the active information-seeking about neonatal jaundice.

Our multivariate analysis revealed that mothers who had a male infant were more likely to be knowledgeable about neonatal jaundice compared with mothers who had a female infant. Health workers generally perform routine jaundice evaluation during birth hospitalisation, and male sex is a risk factor for neonatal jaundice.[37] We speculated that mothers who gave birth to male infants had more opportunities to receive information about neonatal jaundice. A high education level was determinant of knowledge about neonatal jaundice. This finding was consistent with the results of a study from Egypt[13] that found mothers who were university graduates had the highest knowledge scores. This suggested it is necessary for medical staff to provide information about jaundice to mothers with lower education levels. Interestingly, for the mothers without yuesao, the binary logistic regression analysis revealed that good knowledge about jaundice was associated with receiving prior health education on neonatal jaundice from medical staff. However, for the mothers with yuesao, no statistically significant association was detected between prior health education on neonatal jaundice and knowledge towards jaundice. This may be because the yuesao offers a valuable resource for jaundice counselling. Chinese tradition dictates that new mothers stay home and rest for a 'confinement period' of about 1 month (28–42 days) after giving birth, which is thought to facilitate recovery. Previous studies reported that being cared for by a yuesao during this stage can help improve the health of mothers and babies,[38 39] reduce postpartum depression[40 41] and facilitate breast feeding.[42] Therefore, hiring a yuesao to prepare postpartum foods and help with household and childcare tasks has become increasingly popular in many urban families. However, not everyone can afford their services. This suggested it is necessary for hospitals to give mothers pamphlet on neonatal jaundice in which information on recommended websites to seek further information can be obtained. This will help mothers especially those who cannot afford a yuesao.

Our multivariate analysis of practices related to neonatal jaundice revealed that mothers who had received prior education on neonatal jaundice from medical staffs were more likely to have good practices related to jaundice

than other mothers. This finding was consistent with a study conducted in Nigeria[43] that showed mothers who obtained knowledge about neonatal jaundice from medical staff were significantly less likely to self-treat and more likely to seek medical treatment promptly. We also found that mothers with good knowledge about neonatal jaundice were more likely to have good attitudes and practices; this was consistent with the 'knowledge, attitude and practices' (KAP) model,[44] which suggests greater knowledge is the basis for good attitudes and practices.

## Limitations

This study had some limitations. One limitation was that when recruiting mothers, we might have motivated them to learn about neonatal jaundice, which would have improved mothers' awareness of neonatal jaundice before the investigation. However, Chinese traditional culture indicates that mothers need to confine themselves for 1 month after giving birth, so they do not leave their homes during this period. This also made it difficult to recruit mothers after discharge from the hospital. To increase the accessibility of the population, this study recruited mothers in advance during the hospitalisation period after delivery. Another limitation was that our research findings are only representative of mothers' KAP, but for some infants, the main caregivers are other people such as grandmothers, sisters or aunts. The findings of this study are not representative of these individuals.

## CONCLUSION

Overall, mothers' knowledge about jaundice was low, especially regarding causes, danger signs and breast milk jaundice. Active information-seeking behaviour about neonatal jaundice needs to be improved. Therefore, it is recommended that medical staff incorporate information about the causes, danger signs and breast milk jaundice into neonatal jaundice health education programmes and provide reliable websites for mothers to obtain information about neonatal jaundice. This study also showed that the mother's education level was an important factor that is significantly associated with knowledge about jaundice. In addition, mothers receiving care from a yuesao tend to have positive attitudes toward jaundice. Enhancing jaundice-related education programmes targeting mothers with a low education level and no yuesao care is important.

**Author affiliations**
[1]Department of Nursing, Shenzhen Hospital, Southern Medical University, Shenzhen, Guangdong, China
[2]School of Nursing, Southern Medical University, Guangzhou, Guangdong, China
[3]Department of Obstetrics, Shenzhen Hospital of Southern Medical University, Shenzhen, Guangdong, China
[4]Department of Obstetrics, Shen zhen shi bao an qu fu you bao jian yuan, Shenzhen, Guangdong, China

**Acknowledgements** We would like to acknowledge seven neonatologists and my research team for giving feedback to the questionnaire development. Our appreciation also goes to the study participants for generously spending time and providing information in this survey.

**Contributors** YH, LC and XW designed the study. YH, XW, CZ, ZG, FY and JL collected the data. YH, LC and XW analysed the data. YH drafted the manuscript. WC, YH and LC contributed to the interpretation of the results and critical revision of the manuscript for important intellectual content and approved the final version of the manuscript. All authors have read and approved the final manuscript. WC, YH and LC are the study guarantors.

**Funding** This study was supported by grants from Sanming Project of Medicine in ShenZhen, China (SZSM201612018).

**Competing interests** None declared.

**Patient and public involvement** Patients and/or the public were involved in the design, or conduct, or reporting, or dissemination plans of this research. Refer to the Methods section for further details.

**Patient consent for publication** Not applicable.

**Ethics approval** This study involves human participants and was approved by the Ethical Review Committee of Shenzhen Hospital of Southern Medical University (NYSZYYEC20210004). Data were collected from each participant after they received a clear explanation of the purpose and importance of this study and provided informed consent. Participating mothers were informed that participation was voluntary, and they could withdraw from the study at any time or refuse to answer any question. They were also informed they could ask for clarification about any aspect of the study and that the study would not cause harm. Participants did not receive any monetary incentive to participate in this study. All personal information was deidentified and kept securely, and every effort was made to maintain participants' confidentiality. Furthermore, after the investigation, each mother was informed via mobile phone text messages about seeking healthcare from a nearby clinic immediately if any signs of jaundice were identified. Participants gave informed consent to participate in the study before taking part.

**Provenance and peer review** Not commissioned; externally peer reviewed.

**Data availability statement** All data relevant to the study are included in the article or uploaded as supplemental information.

**ORCID iD**
Wenzhi Cai http://orcid.org/0000-0002-2354-5199

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
