## [Reviewer comments · BMJ Open]

ARTICLE DETAILS

TITLE (PROVISIONAL)	Maternal knowledge, attitudes and practices related to neonatal jaundice and associated factors in Shenzhen, China: a facility-based cross-sectional study.
AUTHORS	Huang, Ying; Chen, Ling; Wang, Xiaojiao; Zhao, Chun; Guo, Zonglian; Li, Jue; Yang, Fang; Cai, Wenzhi

VERSION 1 – REVIEW

REVIEWER	Tina Slusher University of Minnesota Academic Health Center, Pediatrics
REVIEW RETURNED	10-Dec-2021

GENERAL COMMENTS	Thank you for doing this study to demonstrate the knowledge, attitudes and practices of your mothers in regard to neonatal jaundice and severe neonatal jaundice or hyperbilirubinemia. The discussion could be strengthened by going into more detail on a few of your points---i.e., is the only reason mothers of male infants were more knowledgeable about jaundice because there is more jaundice in male infants? You could also highlight and discuss why your mothers had better KAP in some areas than has been seen in other countries. Also is G6PD deficiency high in your area of China and if so are there practices that are problematic for neonates who are G6PD deficient that you could have asked about and potentially educated about? While I agree with you that breast milk jaundice is usually benign I would never completely downplay the need for any neonate who is jaundice to be evaluated. Even though it is very rare as pointed out in an old article by Maisels and Newman kernicterus can occur in healthy breastfed infants with no other identified cause for jaundice (Pediatrics 1995). Again, as you correctly point out this should generally be downplayed and only exceedingly rarely should breastfeeding be interrupted (i.e. neonate displaying signs of ABE). Definition of yuesao should be in abstract since it is a term many readers will be unfamiliar with. True misspelled in Table 1 There are many places where there are not spaces between words but that can be corrected at time of publication if accepted. Page 4 Line 44 Run on sentence....kernicterus. They are central.... Page 5 Line 7.evaluation tools or reports.... Page 8 Line 9 Would have been a better question if you had said Breast milk jaundice is generally benign. There are instances where it is not—rare but cannot say absolutely always benign. Of course I know it is impossible to change that now Page 9 Line 11-13neonatal jaundice with only 45.5% of participating mothers demonstrating good knowledge..... Page 9 Line 15about neonatal jaundice. This suggested it is..... Page 9. Lines 38-42 Run on sentence. Having poor knowledge of
---

	breast milk jaundice may also mean mothers discontinue breastfeeding after jaundice occurs. However brief, such discontinuation may jeopardise an infant's ability to return to exclusive breastfeeding, which is unnecessarily harmful to the infant and traumatic for mothers[23]. Page 10 Line 11 Run on sentence..... parenting knowledge. However, they complain that..... Page 10 Lines 29-30 Unclear if this reference belongs here or with the paragraph about healthcare literacy. "This finding was consistent with the results of a study from Egypt[14] that confirmed it is necessary and effective for medical staff to provide information about jaundice to mothers, especially those with lower education levels." Should discuss study limitatons.
--	--

REVIEWER	Ari Satyagraha Lembaga Eijkman
REVIEW RETURNED	17-Dec-2021

GENERAL COMMENTS	Dear Authors, I am a biochemist and molecular biologist and thus I learned quite a lot by reading your manuscript from social science behaviour. I realized the importance of training mothers (especially first time mothers) on neonatal jaundice (NNJ) and as such, your result gives a tremendous input for the health community as a way of prevention strategy on NNJ. However, this study is not new and has been done in many countries and also in China (Ling Zang et al, 2015). What I find interesting is that the title of your manuscript is very similar to that published in BMJ Open as well this year from a study in Ethiopia (Dennis et al, 2021). I have several questions regarding this manuscript:  1. a) Before the survey, you said you interviewed 20 mothers to ensure that the questionnaires can be understood. I wonder how you chose these mothers and whether socially, economically and academically representative to your study subjects? b) Is the questionnaire being reviewed by lay person(s)? 2. In this hospital, when the mothers are normally discharged? 3. Line 1 p. 7 - what is the criteria of "unreasonable responses"? 4. Line 7-15 p. 9 - how do you associate the mothers' lack of knowledge on NNJ with inadequate training of the health worker? Were the 80.6% of mothers with prior education on NNJ trained by health worker? How long ago was the training? Since this will account also to only 45.5% of mothers who actually had good knowledge of NNJ. The fresher the training the better the outcome. 5. From 403 mothers, did any of their newborns experience NNJ? 6. I think previous exposure to NNJ - family history/friends with NNJ history also play a major role in the attitude towards NNJ - not mentioned in this manuscript. I wonder if those with good attitudes have higher education, previous experience/exposure to NNJ. Minor comments: There are many words that are connected together, this could be because of the software problem but I found this throughout the manuscript (eg. p. 5 line 17 diabilityand; line 56 anyaspect), typos: line 46, p. 17 jaundic, Table 2 Ture, Table 5 line 32 neonatec.
--

VERSION 1 – AUTHOR RESPONSE

Reviewer: Dr. Tina Slusher, University of Minnesota Academic Health Center

1. Comment: The discussion could be strengthened by going into more detail on a few of your points---i.e., is the only reason mothers of male infants were more knowledgeable about jaundice because there is more jaundice in male infants?

Reply: Thank you for noting this. After discussion among the authors, we could not offer a better explanation; however, we have revised our explanation of this phenomenon. We have noted that health workers generally perform routine jaundice evaluation during birth hospitalisation. As male sex is a risk factor for neonatal jaundice,^[1] we speculated that mothers who gave birth to male infants had more opportunities to receive information about neonatal jaundice (see the revised *Discussion*, lines 1–4, page 11).

2. Comment: You could also highlight and discuss why your mothers had better KAP in some areas than has been seen in other countries.

Reply: Thank you for your valuable advice. Based on your suggestion, we have highlighted and discussed in the revised manuscript why our mothers had better knowledge in some areas than has been reported in other countries. Because of the large differences in scoring algorithms or items assessing attitudes and practices,^[2-4] it was difficult to compare attitudes and behaviours with other countries (*Discussion*, lines 2–8, page 9).

3. Comment: Also is G6PD deficiency high in your area of China and if so are there practices that are problematic for neonates who are G6PD deficiency that you could have asked about and potentially educated about?

Reply: Thank you for bringing this important point to our attention. First, in Shenzhen, the incidence of G6PD deficiency was reported as 3.54%,^[5] which is higher than the average incidence of G6PD deficiency in China^[6] (0.77%). Second, traditional Chinese medicine is widely used in China to prevent or treat neonatal jaundice.^[7-8] However, neonates with G6PD deficiency that use such remedies may have severe jaundice.^[9] Therefore, mothers whose neonates have G6PD deficiency should be educated about avoiding using traditional Chinese medicine to treat or prevent neonatal jaundice. The questions you raised are valuable, and we have addressed these points in the revised manuscript (*Discussion*, lines 9–14, page 10).

4. Comment: While I agree with you that breast milk jaundice is usually benign I would never completely downplay the need for any neonate who is jaundice to be evaluated. Even though it is very rare as pointed out in an old article by Maisels and Newman kernicterus can occur in healthy breastfed infants with no other identified cause for jaundice (Pediatrics 1995). Again, as you correctly point out this should generally be downplayed and only exceedingly rarely should breastfeeding be interrupted (i.e. neonate displaying signs of ABE).

Reply: Thank you for your comment. We agree with your opinion that we should never completely downplay the need for any neonate who is jaundiced to be evaluated, including those with breast milk jaundice. Based on your comment, we have noted that breast milk jaundice needs routine evaluation in the revised manuscript (*Discussion*, lines 1–9, page 10).

5. **Comment:** Definition of yuesao should be in abstract since it is a term many readers will be unfamiliar with.

Reply: As suggested, we added a definition for yuesao to the abstract (*Abstract*, lines 22, page 2).

6. **Comment:** True misspelled in Table 1

Reply: We apologise for our error. This error was in Table 2, and we have corrected it in the revised Table 2.

7. **Comment:** There are many places where there are not spaces between words but that can be corrected at time of publication if accepted.

Reply: We appreciate this valuable comment. We have carefully checked that missing spaces between words have been corrected in the revised manuscript.

8. **Comment:** Page 4 Line 44 Run on sentence....kernicterus. They are central....

Reply: Thank you for your careful review. We have corrected this issue in the revised manuscript (*Introduction*, lines 25, page 3).

9. **Comment:** Page 5 Line 7.evaluation tools or reports....

Reply: Thank you for your careful review. We have corrected this issue in the revised manuscript (*Introduction*, lines 7, page 4).

10. **Comment:** Page 8 Line 9 Would have been a better question if you had said Breast milk jaundice is generally benign. There are instances where it is not rare but cannot say absolutely always benign. Of course I know it is impossible to change that now.

Reply: Thank you for your comment. The Chinese expression used in that item also meant that breast milk jaundice is usually benign, but it was omitted in the process of writing because of translation problems. Based on your suggestion, a more accurate description has been used in the Discussion section (*Discussion*, lines 2–4, page 10).

11. **Comment:** Page 9 Line 11-13neonatal jaundice with only 45.5% of participating mothers demonstrating good knowledge.....

Reply: Thank you for your careful review. We have corrected this issue in the revised manuscript (*Discussion*, lines 11, page 9).

12. **Comment:** Page 9 Line 15about neonatal jaundice. This suggested it is.....

Reply: Thank you for your careful review. We made major changes in this part of the text based on the opinions of another reviewer. Therefore, the above problem has been resolved (Discussion, lines 12-20, page 9).

13. Comment: Page 9. Lines 38-42 Run on sentence. Having poor knowledge of breast milk jaundice may also mean mothers discontinue breastfeeding after jaundice occurs. However brief, such discontinuation may jeopardise an infant's ability to return to exclusive breastfeeding, which is unnecessarily harmful to the infant and traumatic for mothers[23].

Reply: Thank you for your careful review. We have corrected this point in the revised manuscript (Discussion, line 5, page 10).

14. Comment: Page 10 Line 11 Run on sentence..... parenting knowledge. However, they complain that.....

Reply: We apologise for this grammatical error. We have corrected this issue in the revised manuscript (Discussion, lines 24, page 10).

15. Comment: Page 10 Lines 29-30 Unclear if this reference belongs here or with the paragraph about healthcare literacy. "This finding was consistent with the results of a study from Egypt[14] that confirmed it is necessary and effective for medical staff to provide information about jaundice to mothers, especially those with lower education levels."

Reply: We apologise for our unclear description that resulted in unnecessary misunderstanding. This reference indeed belongs here. This reference shows that mothers who had graduated from university had the highest knowledge score,^[10] which was confirmed in our study. Therefore, we suggested it is necessary for medical staff to provide information about jaundice to mothers with lower education levels (Discussion, lines 6–8, page 10).

16. Comment: Should discuss study limitations.

Reply: Based on your suggestion, we have discussed study limitations in the revised manuscript (Limitations, lines 1–10, page 12).

Specific replies to Reviewer: Dr. Ari Satyagraha, Lembaga Eijkman

1. Comment: However, this study is not new and has been done in many countries and also in China (Ling Zang et al, 2015). What I find interesting is that the title of your manuscript is very similar to that published in BMJ Open as well this year from a study in Ethiopia (Dennis et al, 2021).

Reply: Thank you for this valuable feedback. First, similar investigations have been conducted in other countries or regions (e.g. Ghana,^[11] Accra,^[12] Egypt^[10]). However, no evaluation tools or reports related to maternal knowledge, attitudes and practices about neonatal jaundice are available in China. Regarding the reference you mentioned, we also noted this study when preparing our investigation. It was a prospective study^[13] in which 1036 primiparas were separated randomly into two observation groups

ps: the intervention group received an educational pamphlet, and the control group received routine health education except for information on neonatal jaundice. When infants were 28 days old, a structured questionnaire with 16 items was administered to all first-time mothers to assess their knowledge of neonatal jaundice. It was not a survey of knowledge, attitudes and behaviours regarding neonatal jaundice among mothers. However, maternal jaundice instruction should be given high priority in neonatal jaundice management,^[14] and effective instruction starts with meaningful engagement between hospital staff and mothers.^[15] This highlights that hospital staff need to clarify what mothers know about jaundice and their current attitudes and practices, which will allow health education programmes to target identified gaps. Therefore, our research is valuable and merits publication. We have made appropriate additions to the Introduction to clarify this point (*Introduction, lines 5–6, page 4*).

Second, although the title of our manuscript is similar to that of another study published in *BMJ Open* this year (Dennis et al., 2021), that research differs from our study. Dennis et al.^[16] reported maternal knowledge and attitudes related to neonatal jaundice and knowledge associated factors, whereas we reported maternal knowledge, attitudes and behaviours regarding neonatal jaundice and associated factors.

2. Comment: Before the survey, you said you interviewed 20 mothers to ensure that the questionnaire can be understood. I wonder how you chose these mothers and whether socially, economically and academically representative to your study subjects?

Reply: Thank you for your comments. We apologise for our unclear description. First, 20 mothers were conveniently selected from the same hospital based on the inclusion criteria and exclusion criteria (as clarified in *Data collection tools, lines 18–19, page 5*). Second, these 20 mothers did not statistically significantly differ from the study participants (403 mothers) in terms of social, economic and academic characteristics. The results of our statistical analysis are shown in the following Table

Comparison of sociodemographic data in the 20 mothers and 403 mothers

Variables	Characteristics	403 mothers n (%)	20 mothers n (%)	χ^2	p
Age (years)	19–27	114 (28.3)	8 (40)	1.391	0.497
	28–32	197 (48.9)	9 (45)		
	33–45	92 (22.8)	3 (15)		
Blood group	O	145 (36.0)	8 (40)	0.439	0.956
	A	121 (30.0)	5 (25)		
	B	106 (26.3)	6 (30)		
	AB	31 (7.7)	1 (5)		

	High school and below	68 (16.9)	4 (20)		
Education level	University	310 (76.9)	14 (70)	1.212	0.534
	Postgraduate and above	25 (6.2)	2 (10)		
	Employed	267 (66.3)	11 (55)		
Occupation	Self-employed	40 (9.9)	3 (15)	3.574	0.255
	Housewife	82 (20.3)	4 (20)		
	Others	14 (3.5)	2 (10)		
	≤5000	50 (12.4)	4 (20)		
Average family monthly income (RMB)	5001~10000	154 (38.2)	7 (35)		
	10001~20000	125 (31)	5 (25)	2.186	0.708
	20001~30000	35 (8.7)	1 (5)		
	≥30001	39 (9.7)	3 (15)		
	≤10	52 (12.9)	2 (10)		
Time from place of residence to the delivery hospital (minutes)	10~30	212 (52.6)	11 (55)	0.760	0.861
	30~60	126 (31.3)	6 (30)		
	≥60	13 (3.2)	1 (5)		

3. Comment: Is the questionnaire being reviewed by lay person(s)?

Reply: By 'lay person(s)', were you referring to the mothers? If so, then yes, the questionnaire was reviewed by lay persons. Following the review by the expert panel, nine mothers whose babies had experienced neonatal jaundice were conveniently recruited to provide input on the importance and clarity of the questionnaire items. These mothers were asked to suggest alternative wording for existing items and identify any items for deletion and addition. Based on their review, we modified 'Have you ever learned about neonatal jaundice' to 'Prior health education on neonatal jaundice', 'G6PD deficiency' to 'Broad bean disease' and 'Serum total bilirubin is the gold standard for diagnosing neonatal jaundice' to 'Blood test is the gold standard for diagnosing neonatal jaundice'. No items were suggested for deletion or addition. We have added this information to the manuscript (*Data collection tools*, lines 11–17, page 5).

4. Comment: In this hospital, when the mothers are normally discharged?

Reply: In the study hospital, mothers and their newborns are normally discharged within 48~72 hours after vaginal delivery or 96~120 hours after caesarean delivery. We have described this in detail in the revised manuscript to clarify (*Abstract, lines 8–9, page 2*).

5. Comment: Line 1 p. 7 - what is the criteria of "unreasonable responses"?

Reply: We apologise for our unclear description. One participant chose the first option for each question and was not included in the analysis. We have described this in the revised manuscript (*Data collection procedures, lines 13–14, page 6*).

6. Comment: Line 7-15 p. 9 - how do you associate the mothers' lack of knowledge on neonatal jaundice with inadequate training of the health worker? Were the 80.6% of mothers with prior education on neonatal jaundice trained by health worker? How long ago was the training? Since this will account also to only 45.5% of mothers who actually had good knowledge of neonatal jaundice. The fresher the training the better the outcome.

Reply: We apologise for our unclear description. First, yes, the 80.6% mothers with prior education on neonatal jaundice were trained by health workers. Second, they were trained on the day of normal discharge from the delivery hospital (48~72 hours after vaginal delivery or 96~120 hours after caesarean delivery). About 5 days after discharge from the hospital, we sent the questionnaire link to participating mothers (*Results, lines 9–11, page 7*). Third, thank you for the reminder. Our previous explanation for the mothers' lack of knowledge about neonatal jaundice was associated with inadequate training of health workers was unclear. Therefore, we revised this explanation. The poor knowledge may be attributable to the fact that there was a gap of nearly a week between the time they received health education and the time of our investigation; mothers might have forgotten the content of health education. In addition, medical staff only provided post-discharge monitoring and follow-up instruction and did not include neonatal jaundice knowledge that related to the questionnaire when conducting health education. Another factor that might have contributed to the comparative ineffectiveness of postnatal instruction was that the unique environment was absent that mothers received health education about jaundice from health workers in a single setting, which combined a lecture, demonstration and interactive discussion, as there is generally a rush to discharge mothers from birthing centres. We have noted this in the revised manuscript (*Discussion, lines 12–20, page 9*).

7. Comment: From 403 mothers, did any of their newborns experience neonatal jaundice?

Reply: We apologise for our unclear description. Among the 403 mothers, 113 (28%) reported their current child was admitted to hospital for treatment due to jaundice after discharge, and 56 (13.9%) reported that a previous child had a history of neonatal jaundice. We have noted this in the revised manuscript (*Results, lines 11–13, page 7*).

8. Comment: I think previous exposure to neonatal jaundice - family history/friends with neonatal jaundice history also play a major role in the attitude towards neonatal jaundice - not mentioned in this manuscript. I wonder if those with good attitudes have higher education, previous experience/exposure to neonatal jaundice.

Reply: Thank you for your comments.

First, we omitted to include previous exposure to neonatal jaundice (family history/friends with neonatal jaundice history) in the baseline characteristics of participating mothers. Based on your suggestion, we attempted to recontact the 403 mothers; 373 mothers were successfully contacted and included in an analysis of previous exposure to neonatal jaundice. However, the chi-square tests revealed that attitude had no significant correlation to family history/friends with neonatal jaundice history ($\chi^2=0.100$, $P=0.752$). We have added this point to the revised manuscript (Table 1).

Second, the chi-square tests (bivariable analyses) showed that previous experience/exposure to neonatal jaundice had a non-significant impact on attitude. However, a positive attitude towards jaundice was associated with higher education ($\chi^2=9.80$, $P=0.007$). These chi-square test results are shown in the following Table. Our multivariable analysis revealed that a positive attitude towards jaundice was associated with being cared for by a 'yuesao' (matron specialised in maternal and newborn care) and good knowledge about jaundice, and had a non-statistically significant association with education level. Overall, there was no statistically significant association between education level, previous experience/exposure to neonatal jaundice and attitude toward neonatal jaundice. We had performed chi-square tests (bivariable analyses) in the original manuscript, but because of limited space, did not present the results of chi-square tests (bivariable analyses) for maternal knowledge, attitudes and practices related to neonatal jaundice. We have added these results as supplementary information (Supplementary table 1)

Variables associated with maternal attitude related to neonatal jaundice (N=403)

Variables	Attitude		χ^2	P
	Poor (%)	Good (%)		
Education level				
High school and below	40 (58.8)	28 (41.2)	9.80	0.007
University	122 (39.4)	188 (60.6)		
Postgraduate and above	8 (32)	17 (68)		
Previous experience/exposure to neonatal jaundice				
Prior health education on neonatal jaundice				
Yes	151 (41.8)	210 (58.2)	0.179	0.672
No	19 (45.2)	23 (54.8)		
Previous child history of neonatal jaundice				
Yes	24 (42.9)	32 (57.1)	0.140	0.708
No	146 (42.1)	210 (57.9)		

Current child admitted to the hospital for treatment for jaundice after discharge

Yes	46 (40.7)	67 (59.3)	0.012	0.912
No	124 (42.8)	233 (57.2)		

Family history/friends with neonatal jaundice history*

Yes	20 (44.4)	25 (55.6)	0.100	0.752
No	154 (47.0)	174 (53.0)		

Note: * N=373

9. Comment: Minor comments:

There are many words that are connected together, this could be because of the software problem but I found this throughout the manuscript (eg. p. 5 line 17 disabilityand; line 56 anyaspect), typos: line 46 , p. 17 jaundic, Table 2 Ture, Table 5 line 32 neonatec.

Reply: We apologise for this error, and have corrected this through out the manuscript.

VERSION 2 – REVIEW

REVIEWER	Tina Slusher University of Minnesota Academic Health Center, Pediatrics
REVIEW RETURNED	11-Mar-2022

GENERAL COMMENTS	.Much improved. Message now clear. Thanks for the revisions
---

REVIEWER	Ari Satyagraha Lembaga Eijkman
REVIEW RETURNED	04-Mar-2022

GENERAL COMMENTS	Dear Authors, I have read your revised manuscript and it is more clear now than before. I have commented on small typos in the abstract. I have a few comments/questions: 1. Is having Yuesao limited to those of the upper class or everyone can afford their services? Because if not everyone can afford their services, it'll affect only those with certain salary range and education. Thus, another comparison should be done to differentiate those without yuesao and with yuesao in correlation to education, salary in terms of attitude and knowledge. 2. I suggest that the authors should recommend hospitals to give mothers pamphlet on neonatal jaundice in which informations on recommended websites to seek further information can be obtained. This will help mothers especially those who cannot afford a yuesao. The reviewer provided a marked copy with additional comments. Please contact the publisher for full details.
---

VERSION 2 – AUTHOR RESPONSE

Dr. Ari Satyagraha, Lembaga Eijkman

1. Comment: I have commented on small typos in the abstract.

Reply: Thank you for your careful review. We have corrected this issue in the revised manuscript (Abstract, lines 8–9, page 2).

2. Comment: Is having Yuesao limited to those of the upper class or everyone can afford their services? Because if not everyone can afford their services, it'll affect only those with certain salary range and education. Thus, another comparison should be done to differentiate those without yuesao and with yuesao in correlation to education, salary in terms of attitude and knowledge.

Reply: Thank you for noting this. Yes indeed, not everyone can afford to have Yuesao services. Our survey data found that the mothers with yuesao was associated with a high level of education ($\chi^2=20.025$, $P=0.000$) and a high level of salary ($\chi^2=30.803$, $P=0.000$) than those who did not have yuesao. These chi-square test results are shown in the following Table. Based on your suggestion, we have done a comparison to differentiate those without yuesao and with yuesao in correlation to education, salary in terms of attitude and knowledge in our revised manuscript. We found that despite the mothers without or with yuesao, there was no statistically significant association between salary in terms of attitude and knowledge about jaundice.

When considering education, for the mothers who having yuesao, the binary logistic regression analysis revealed that positive attitude about jaundice was associated with a high level of education. (i.e., College and undergraduate course; odds ratio [OR]=7.683, 95% confidence interval [CI]: 1.583–37.297, $P=0.011$). However, for the mothers without yuesao, no statistically significant association was detected between education level and attitude towards jaundice. These results are shown Supplementary table 2 and Supplementary table 3. Based on the results, we revised the Methods, Results and Discussion of the manuscript. (Methods, lines 1–4, page 7; Results, lines 13–15, page 12; Discussion, lines 8–21, page 17).

3. Comment: I suggest that the authors should recommend hospitals to give mothers pamphlet on neonatal jaundice in which information on recommended websites to seek further information can be obtained. This will help mothers especially those who cannot afford a yuesao.

Reply: Thank you for your valuable advice. Based on your suggestion, we have added that hospitals should give mothers pamphlet on neonatal jaundice in which information on recommended websites to seek further information can be obtained, which will help mothers especially those who cannot afford a yuesao in the revised manuscript. (Discussion, lines 18–21, page 17).

Table. The mothers without yuesao and with yuesao in correlation to education, salary (N=403)

Variables	without yuesao(%)	with yuesao(%)	χ^2	P
Education level				
High school and below	57(21.5)	11(8)	20.025	0.000
University	199(75.1)	111(80.4)		
Postgraduate and above	9(3.4)	16(11.6)		
Average family monthly income, RMB				
≤5000	43(16.2)	7(5.1)	30.803	0.000
5001~10000	112(42.3)	42(30.4)		
10001~20000	78(29.4)	47(34.1)		
20001~30000	13(4.9)	22(15.9)		
≥30001	19(7.2)	20(14.5)